# Enhancing Biochar Impact on the Mechanical Properties of Cement-Based Mortar: An Optimization Study Using Response Surface Methodology for Particle Size and Content

Zhongrui Zhou [1], Junsong Wang [1], Kanghao Tan [2] and Yifei Chen [3,*]

1   State Key Laboratory of Subtropical Building Science, School of Architecture, South China University of Technology, Guangzhou 510640, China
2   School of Materials Science and Engineering, South China University of Technology, Guangzhou 510641, China
3   School of Civil Engineering and Architecture, Guangxi University, Nanning 530004, China
*   Correspondence: lygcyfm@126.com

**Abstract:** The utilization of agricultural waste, specifically biochar (BC), as an alternative material to conventional Portland cement offers substantial potential for enhancing sustainability within the construction industry. This study investigates how variations in BC particle size and content affect the properties of cement mortar using Response Surface Methodology (RSM). By manipulating BC's content and particle size in the mortar mixture and analyzing the data with RSM, this study establishes response surface models to predict the relationship between BC characteristics and cement mortar strength. The results demonstrate that the optimal combination for enhancing the mechanical performance of the mortar is achieved when BC particles have a median particle diameter of 51.08 μm and a content of 2.69% of the mixture. Additionally, utilizing scanning electron microscopy (SEM), it is revealed that BC serves as a nucleation site for cement hydration, thereby inducing a more compact and dense microstructure within the cement mortar. Furthermore, BC particles contribute to enhancing the interfacial transition zone between the cement paste and aggregate, leading to increased compressive strength and fracture toughness of the mortar while simultaneously curbing crack propagation.

**Keywords:** biochar; response surface method; particle size; mechanical properties; microstructure

## 1. Introduction

Cement, a crucial foundational material, plays a pivotal role in both social and economic development [1,2]. In 2021, global cement production reached approximately 4.28 billion tons [3]. This significant production, however, comes at a cost: cement production relies heavily on calcareous raw materials and fossil fuels, resulting in an average $CO_2$ emission of approximately 610 kg per unit produced [4]. Consequently, it can be calculated that in 2021, the $CO_2$ emissions from the global cement industry amounted to nearly 2.6 billion tons, representing approximately 7% of the total $CO_2$ emissions associated with human production activities. Considering the economic growth trajectories of many countries, cement production still possesses significant growth potential [5]. Consequently, the proportion of carbon emissions stemming from cement production is expected to rise in the coming years, further contributing to the overall carbon emissions landscape. This underscores the critical importance of carbon emission reduction, not only for promoting the green and sustainable progression of the cement industry but also both as an opportunity and a challenge that the sector must proactively address.

The development of carbon capture, utilization, and storage (CCUS) technology holds significant promise for reducing carbon emissions in the global cement industry [6]. One particularly viable approach involves incorporating pulverized biochar (BC) as an additive

in cementitious materials [7–9]. This method offers a means to diminish the net $CO_2$ emissions associated with these materials by sequestering stable carbon within them, all while preserving the integral properties of cement composites. BC represents a porous carbonaceous material obtained through the pyrolysis of solid biomass, such as animal waste, food waste, manure, and other agricultural sources, at low temperatures (<700 °C) in the absence of oxygen or under controlled oxygen limitations [10,11]. The addition of BC to cement-based material represents a sustainable solution for reducing $CO_2$ emissions, both by curbing cement production and by facilitating the recycling of waste biomass [12–14].

Recently, BC has gained attention as a potential substitute for cement in cementitious materials. This is due to its ability to enhance their mechanical, thermal, and environmental properties. In general, BC increases the compressive strength of cementitious materials when it is incorporated at lower levels (1–3%). For instance, Gupta [15] found that the compressive strength of cementitious materials doped with 1% BC increased by about 10% compared to the control group. However, when higher amounts of BC (5–30%) are added, it has the opposite effect, diminishing the compressive strength of cementitious materials. According to ref. [16], incorporating 5–10% biochar into cementitious materials resulted in a notable decline in compressive strength, with reductions of around 10% to 15% when compared to the control group. In a more extreme case, as reported by Ref [17], an 8% BC inclusion in cementitious materials led to a substantial 50% decrease in compressive strength in comparison to the control group. Beyond its impact on compressive strength, BC can also influence other mechanical properties of cementitious materials, such as flexural strength. According to Ref [18], the addition of 3% BC to cementitious materials results in an approximate 10% increase in bending strength compared to the control. Likewise, both Gupta et al. [15] and Wang et al. [19] have reported that even a mere 1% BC inclusion results in an approximate 0.2 MPa increase in flexural strength compared to the control group.

Previous research has extensively demonstrated the beneficial impact of incorporating BC as a filler or reinforcing agent in cementitious composites. It enhances the mechanical properties, durability, and environmental sustainability of these materials. However, while BC has shown excellent promise as an auxiliary cementitious material for improving composite properties, limited attention has been given to the influence of BC particle size on cement-based materials. The particle size of BC is intricately linked to its raw material characteristics and the conditions during pyrolysis. Various pyrolysis methods yield BC particles of varying sizes. For instance, fast pyrolysis typically employs smaller feedstock particles to produce finer BC particles, with an average size of 421 μm [20]. In contrast, slow pyrolysis can utilize larger feedstock particles, resulting in coarser BC exceeding 1 cm [21]. To meet specific application requirements, BC produced via slow pyrolysis is typically subjected to milling and sieving, resulting in particle sizes ranging from 0.044 to 20 mm [22]. BC with a finer particle size boasts a larger specific surface area and an increased number of active sites, making it advantageous for applications such as soil remediation and heavy metal adsorption [23–25]. However, it is essential to determine whether fine particle size BC similarly benefits cementitious materials. Some literature suggests that fine-size BC can enhance cement hydration, pore filling, and, subsequently, the strength and durability of cement-based materials [7,26]. Conversely, certain studies indicate that fine-size BC may elevate the viscosity and rheology of cement paste, potentially reducing its workability [26]. Moreover, it is worth noting that the production of finer particle-size BC demands more energy, potentially negating its environmental advantages in terms of reduced carbon emissions. Currently, most studies focus on investigating the impact of BC characteristics (content and particle size) on the properties of cement-based materials through the manipulation of individual variables. However, research on the optimization of BC-cement mortar properties remains limited. The complex interplay between various factors makes it challenging to identify the optimal combination of these factors using a stepwise experimental design approach. These gaps pose challenges to the development of BC-cement composites with superior performance and

hinder their wider application in engineering practice. Response Surface Methodology (RSM) presents a comprehensive experimental design and mathematical modeling optimization approach. Through testing representative local points, this method enables the establishment of functional relationships between each factor and the overall fitting results. It allows for the determination of the optimal values for each factor and facilitates the study of the interactions among multiple factors affecting the desired response value within a global context.

To do so, this study employed the RSM to analyze and optimize the influence of the interaction between BC content and particle size on the mechanical properties, specifically compressive strength and flexural strength, of the cement-based mortar. Additionally, a quadratic regression model was established to accurately characterize the relationship between the mechanical properties and the aforementioned factors. The validity of this model was rigorously assessed through significance tests, analysis of variance (ANOVA), and correlation coefficient analysis. Through this comprehensive approach, the study successfully determined the optimal combination scheme and identified the maximum achievable response value for achieving superior mechanical properties. The notable contribution of this research lies in presenting an effective method for optimizing the mechanical properties of BC-mortar, which can significantly impact the future development of low-carbon, efficient, and sustainable building materials. The findings of this study can serve as a valuable reference for further advancements in this field.

## 2. Materials and Methods

### 2.1. Materials

#### 2.1.1. Cement and Aggregates

This study employs the commercial standard P.O.42.5 type Portland cement manufactured by the Conch brand. The detailed chemical compositions and particle size distribution (PSD) of the cement particles are shown in Table 1 and Figure 1. It is noteworthy that the PSD of the cement spans from 1 to 100 μm, and the median particle diameter ($D_{50}$) is approximately 10 μm.

**Table 1.** Physical properties and chemical composition of cement.

| CaO | SiO$_2$ | Al$_2$O$_3$ | Fe$_2$O$_3$ | MgO | Na$_2$O and K$_2$O | SO$_3$ | P$_2$O$_5$ | LOI | BET Surface Area (m$^2$ g$^{-1}$) |
|---|---|---|---|---|---|---|---|---|---|
| 62.46 | 21.60 | 5.79 | 3.95 | 0.73 | 0.54 | 3.59 | 0.17 | 1.17 | 350 |

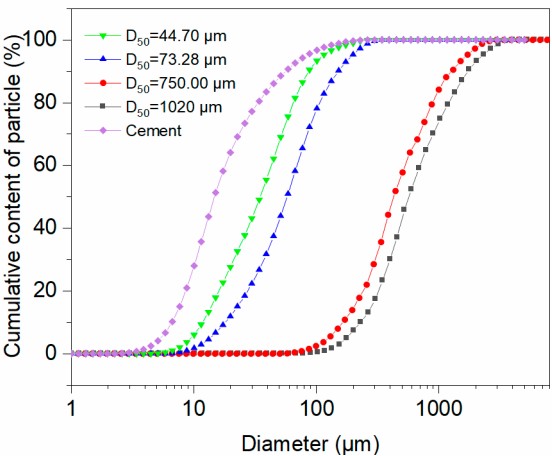

**Figure 1.** PSD of cement particles and the pulverized BC with four different particle sizes.

According to the ISO standard, the constituents comprising the fine aggregate encompass naturally occurring river sand, characterized by a fineness modulus of 2.5. This sand

materializes with an observed apparent density of approximately 2500 kg/m$^3$, concomitant with a quantified bulk density of 1187 kg/m$^3$, and a recorded moisture content of 0.59%.

### 2.1.2. BC Preparation

The waste wood was the first precision-cut material to achieve specific dimensions: 2 cm in length, 2 cm in width, and a thickness of 1.5 mm. Following this, it underwent pyrolysis in a muffle furnace at 500 °C for 2 h, resulting in the creation of BC. Once the pyrolysis phase was completed, the BC was extracted from the muffle furnace. Subsequently, an automatic grinding machine (YB-2500A) with 3000 RPM, purchased from Yongkang City Sufeng Industry and Trade Co., Ltd., Yongkang, China, was used for the grinding process, aiming to reduce the particle size to a range between 10 and 300 mesh, with grinding durations of 20 s, 30 s, 1 min, and 2 min, respectively.

The pulverized BC was then sieved to obtain four different particle sizes, utilizing various mesh sizes: 300 mesh, 200 mesh, 100 mesh, and 10 mesh. To determine the central tendency, the $D_{50}$ of the pulverized BC was calculated using a laser diffraction particle size analyzer (Partica LA-960 A, from HORIBA, Ltd. in Kyoto, Japan). The analysis revealed the particle size distribution, highlighting notable fractions: $D_{50}$ = 44.70 μm, 73.28 μm, 750.00 μm, and 1020 μm, as illustrated in Figure 1. Evidently, it is discernible that the $D_{50}$ of BC exhibits a diminishing trend concomitant with escalating grinding duration. However, it is noteworthy that the grinding efficacy manifests a superiority for coarser particles (>100 μm) as opposed to finer particles (<100 μm). At a grinding duration of 2 min, the $D_{50}$ approximates 44.70 μm, concomitant with a specific surface area akin to that of cement. This implies that a reduced particle size augments the interfacial contact area between BC and cement, thereby instigating their chemical interplay [27]. Furthermore, it mitigates BC agglomeration within the cementitious matrix, thereby ensuring a more homogeneous dispersion.

The SEM images of BC particles at 2000× magnification are shown in Figure 2. As depicted in the SEM images, the irregular honeycomb pore structure on the surface of BC particles can be attributed to the natural porosity of the raw material and the release of numerous volatile gases during the pyrolysis process. The particle size ranges from a few microns to several dozen microns, resulting in a substantial number of micropores on the BC particle surface. This characteristic endows biochar with an enhanced specific surface area, creating an ideal environment for substance adsorption. The elemental composition analysis of BC particles was conducted using an energy dispersive spectrometer (EDS), and the results are presented in Table 2. The findings revealed no considerable variation in chemical composition among different BC particle sizes. Notably, the O/C ratio of BC recorded at 0.12 attests to the BC's commendable stability and durability [28,29]. This characteristic is pivotal when incorporating BC into cement mortar as an auxiliary cement material, as it assures long-term stability.

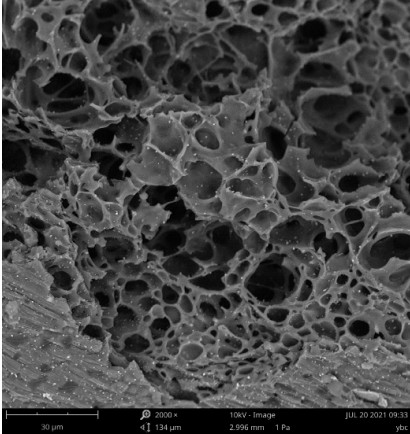

**Figure 2.** SEM images of biochar samples at 2000× magnification.

**Table 2.** Chemical composition and physical properties of BC.

| Grinding Time | $D_{50}$/µm | Ultimate Analysis (wt% daf) | | | | | | | | BET Surface Area (m$^2$ g$^{-1}$) |
|---|---|---|---|---|---|---|---|---|---|---|
| | | C | O | N | Al | Ca | Mg | K | Si | |
| 2 min | 44.70 | 72.60 | 8.96 | 1.78 | 1.89 | 0.51 | 0.30 | 0.30 | 0.40 | 617.2 |
| 1 min | 73.28 | 72.68 | 9.20 | 1.81 | 1.72 | 0.55 | 0.31 | 0.19 | 0.39 | 605.2 |
| 30 s | 750 | 72.68 | 9.21 | 1.79 | 1.89 | 0.54 | 0.34 | 0.25 | 0.40 | 518.9 |
| 20 s | 1020 | 71.10 | 9.21 | 1.78 | 1.89 | 0.50 | 0.29 | 0.21 | 0.40 | 336.9 |

## 2.2. Experimental Design

The response surface method (RSM), a multivariate nonlinear regression technique, optimizes experimental factors and response variables. This integration blends Analysis of Variance (ANOVA) and quadratic polynomial regression models to achieve structured outcomes. In this study, the investigation focused on the effect of BC content ("A") and particle size ("B") on the 7-day and 28-day compressive and flexural strengths of cement-based mortars. This study used the optimal (custom) design module of RSM in Design Expert 12.0 to create a matrix containing 16 samples. This adaptable approach customizes models, categorical factors, and constrained domains. Detailed information regarding classifications and codes is shown in Table 3.

**Table 3.** Levels of independent variables in the stabilization experiments based on RSM.

| Symbol | Factor | Unit | Type | Levels | L [1] | L [2] | L [3] | L [4] |
|---|---|---|---|---|---|---|---|---|
| A | BC content | % | Discrete | 4 | 1 | 3 | 5 | 10 |
| B | BC particle size | µm | Discrete | 4 | 44.21 | 73.3 | 750 | 1020 |

## 2.3. Specimen Preparation

The water-to-binder ($w/b$) ratio in the BC mortar, as outlined by GB/T 17671-2021 [30], is fixed at 0.5. The cement-to-standard sand mass ratio was consistently maintained at 1:3. The amounts of BC incorporated were 1%, 3%, 5%, and 10% by weight of cement, respectively. The mix proportions of the mortar mixtures are given in Table 4. Each pulverized BC sample is intimately blended with cement for ten minutes and then progressively incorporated into the cement mixture. The mortar mixtures were cast into cubic molds with a size of 50 mm × 50 mm × 50 mm for the compressive strength test and rectangular molds with a size of 40 × 40 × 160 mm for the flexural strength test. Voids are eliminated through compaction on a vibrating table. Subsequently, sets of three specimens are placed within a controlled environment chamber with a constant temperature of 20 °C and 98% relative humidity for uniform curing conditions. After an initial 24 h curing period, the specimens are demolded and cured in water at room temperature for 6 and 27 days.

**Table 4.** Mix proportions of mortar mixtures.

| Sample | $w/b$ | Cement (g) | Sand (g) | BC (g) | Water (g) | Remarks Median Diameter of Pulverized BC ($D_{50}$) |
|---|---|---|---|---|---|---|
| Control mortar | 0.5 | 450 | 1350 | 0 | 225 | Mortar without BC |
| BCM-1 | 0.5 | 445.5 | 1350 | 4.5 | 225 | Mortar with different median |
| BCM-3 | 0.5 | 436.5 | 1350 | 13.5 | 225 | diameter of pulverized BC ($D_{50}$ = |
| BCM-5 | 0.5 | 427.5 | 1350 | 22.5 | 225 | 44.70 µm; $D_{50}$ = 73.28 µm; $D_{50}$ = |
| BCM-10 | 0.5 | 405.0 | 1350 | 45.0 | 225 | 750 µm; $D_{50}$ = 1020 µm) |

## 2.4. Methods

### 2.4.1. Mechanical Properties Test

Compressive strength and flexural strength measurements were conducted after the completion of 7 and 28 days of curing. The cube sample (50 × 50 × 50 mm$^3$) was determined

for the test of compressive strength according to ASTM C109 [31]. Flexural strength was tested using a three-point loading method on a prism sample with a size of $40 \times 40 \times 160$ mm$^3$ based on ASTM C 348 [32]. Three samples were tested, and the average strength has been reported.

### 2.4.2. Microscopic Morphology Test

The central part of the fractured specimen in the 28-day compressive strength test was selected for SEM analysis. The specimen was first immersed in anhydrous ethanol to stop hydration and then vacuum-dried at 60 °C for 24 h. Next, sheet-like samples with an area of less than 1 cm$^2$ were cut from the specimen and attached to the sample stage using conductive adhesive. Because the specimens were non-conductive, they underwent ion sputtering for gold coating in an ion-sputtering device before being placed in the scanning electron microscope. The SEM was performed using a Hitachi SU8220 scanning electron microscope with an acceleration voltage of 5 kV.

## 3. Results

### 3.1. Experimental Results

Table 5 shows the results of the optimal (custom) design method and the corresponding responses. Further insights into the experimental findings regarding the compressive and flexural strengths of 16 distinct BC cement mortar blocks are illustrated in Figure 3a,b. Based on the mechanical attributes observed after a 28-day curing period, it is evident that among the trials, Run 12 exhibits the most noteworthy compressive and flexural strengths, registering at 37.81 MPa and 6.77 MPa, respectively. This observation aligns with the research conducted by Gupta et al. [33], wherein it was established that the incorporation of 1–2% by weight of biochar yields a notable enhancement in the initial compressive strength of cement mortar. Conversely, runs 6, 15, and 16 have the lowest compressive strength of 33.84 MPa. Run 15 has the lowest flexural strength of 5.36 MPa, significantly trailing behind the control group strength of 6.82 MPa. Moreover, the significant improvements are not limited to Run 12; other samples, such as Runs 2, 3, and 8, also show better performance in both compressive and flexural strengths. In conclusion, the results highlight the positive effect of adding 1–3% BC as a cement substitute on the mechanical properties of mortar samples. However, it is imperative to acknowledge that the optimization of the mechanical properties of BC-mortar necessitates a holistic consideration of both BC content and particle size.

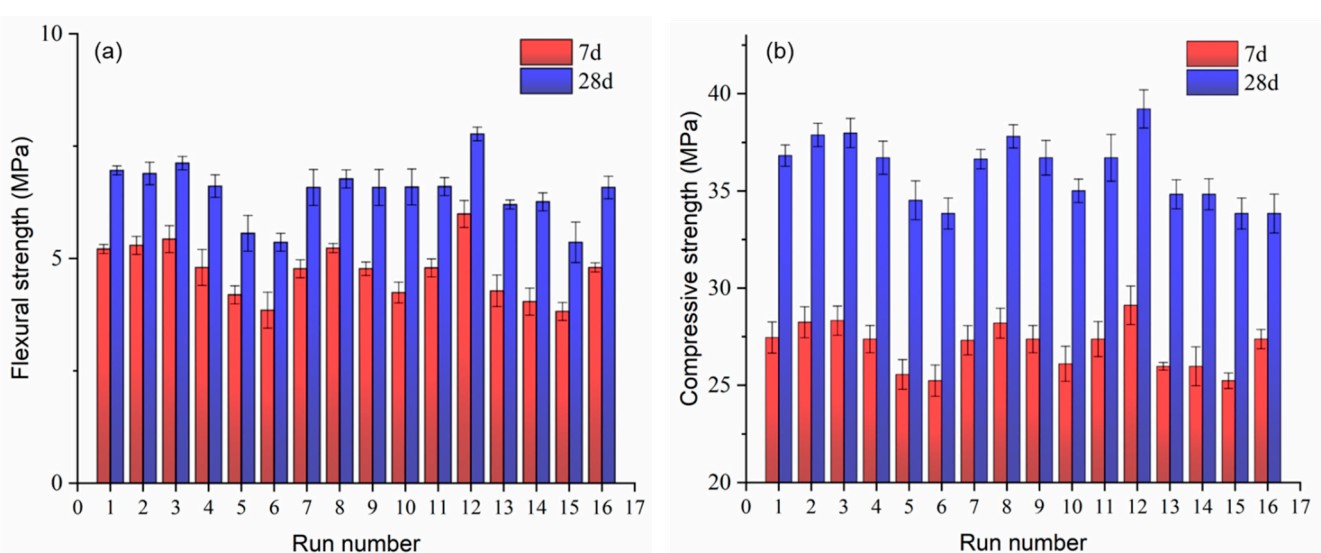

**Figure 3.** Comparative analysis of mechanical property test results: (**a**) flexural strength; (**b**) compressive strength.

**Table 5.** Designed experiments and responses according to RSM.

| Run | Factor | | Response | | | |
|---|---|---|---|---|---|---|
| | A/(%) | B/µm | Compressive Strength (MPa) | | Flexural Strength (MPa) | |
| | | | 7 d ($Y_1$) | 28 d ($Y_2$) | 7 d ($Y_3$) | 28 d ($Y_4$) |
| | Control group | | 26.40 | 35.40 | 4.58 | 6.82 |
| 1 | 5 | 73.3 | 27.46 | 36.82 | 5.21 | 6.96 |
| 2 | 1 | 750 | 28.25 | 37.88 | 5.29 | 6.89 |
| 3 | 1 | 73.3 | 28.33 | 37.98 | 5.43 | 7.12 |
| 4 | 5 | 750 | 27.38 | 36.71 | 4.80 | 6.61 |
| 5 | 10 | 44.21 | 25.56 | 34.52 | 4.19 | 5.56 |
| 6 | 10 | 750 | 25.24 | 33.84 | 3.85 | 5.36 |
| 7 | 5 | 1020 | 27.32 | 36.64 | 4.77 | 6.58 |
| 8 | 1 | 1020 | 28.20 | 37.81 | 5.23 | 6.77 |
| 9 | 5 | 750 | 27.38 | 36.71 | 4.77 | 6.58 |
| 10 | 10 | 73.3 | 26.11 | 35.01 | 4.24 | 6.59 |
| 11 | 5 | 750 | 27.38 | 36.71 | 4.79 | 6.60 |
| 12 | 3 | 44.21 | 29.12 | 39.22 | 5.99 | 7.77 |
| 13 | 5 | 1020 | 26.60 | 34.83 | 4.28 | 6.20 |
| 14 | 10 | 1020 | 25.98 | 34.40 | 4.04 | 6.26 |
| 15 | 10 | 750 | 25.24 | 33.84 | 3.82 | 5.36 |
| 16 | 5 | 750 | 27.38 | 33.21 | 4.80 | 6.58 |

### 3.2. Mechanical Properties Analysis

Figure 4 shows the interaction response surfaces of BC content and particle size on the 7-day and 28-day compressive strengths of cement mortar. The visual representation elucidates that adding BC to cement mortar can increase its compressive strength, depending on the content and particle size of BC. However, this reinforcing effect wanes as the curing time of the cement mortar increases. The two-dimensional contours extend further insight into the interconnected dynamics between BC content and particle size. By maintaining a constant particle size for BC while varying its content within the range of 1% to 5%, the compressive strength of the cement mortar specimens exhibits a minor yet discernible enhancement. Of particular significance, the cement mortar achieves its peak compressive strength of approximately 29 MPa on the 7th day and further increases to 39 MPa on the 28th day when the BC content is set at 5% and the particle size measures 44.21 µm. However, adding more than 5% BC reduces the strength significantly. In contrast, under constant BC content conditions, an increase in BC particle size leads to a converse impact on the compressive strength of mortar specimens. For instance, when the BC content is maintained at 3%, elevating the particle size from 44.21 µm to 1020 µm precipitates a reduction of 0.4 MPa and 0.6 MPa in compressive strength at 7 and 28 days, respectively. This suggests that larger BC particles may weaken the bond between BC and cement, affecting the mechanical properties of the mortar.

Figure 5 shows the response surface of the interactions between BC content and particle size on the 7-day and 28-day flexural strengths of mortar samples. It shows that the optimal mix of BC-contained mortar has less than 3% BC and less than 100 micron particle size, as indicated by the red region in Figure 5a. It also reveals that a 1–5% BC content significantly bolsters cement mortar strength, particularly for smaller particle sizes (Figure 5b). Comparing Figures 4a and 5a, it becomes evident that the response surface in Figure 5a exhibits a more pronounced curvature. This observation implies that particle size and content exert a more significant influence on the flexural strength of cement mortar as opposed to its compressive strength.

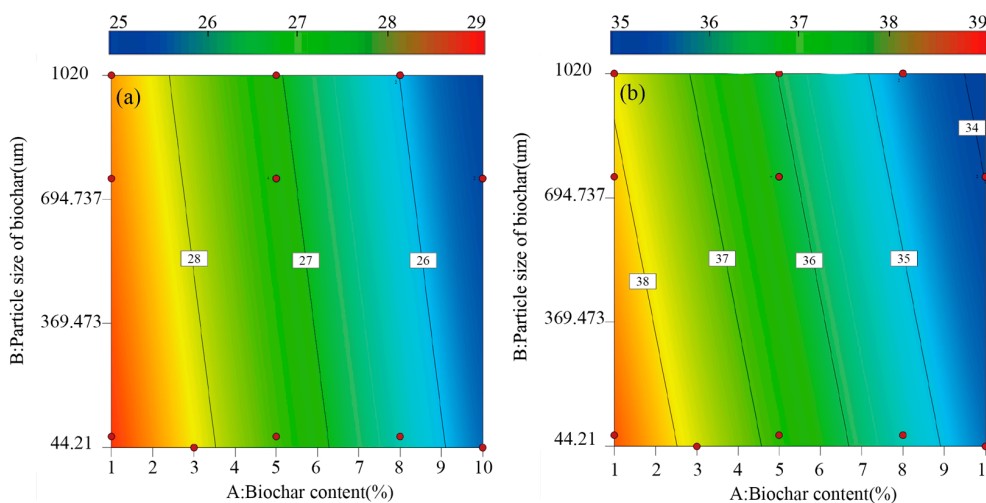

**Figure 4.** RSM analysis of compressive strength: (**a**) 2-D plots for 7-day compressive strength; (**b**) 2D plots for 28-day compressive strength (Note: The red dot represents the actual flexural strength value).

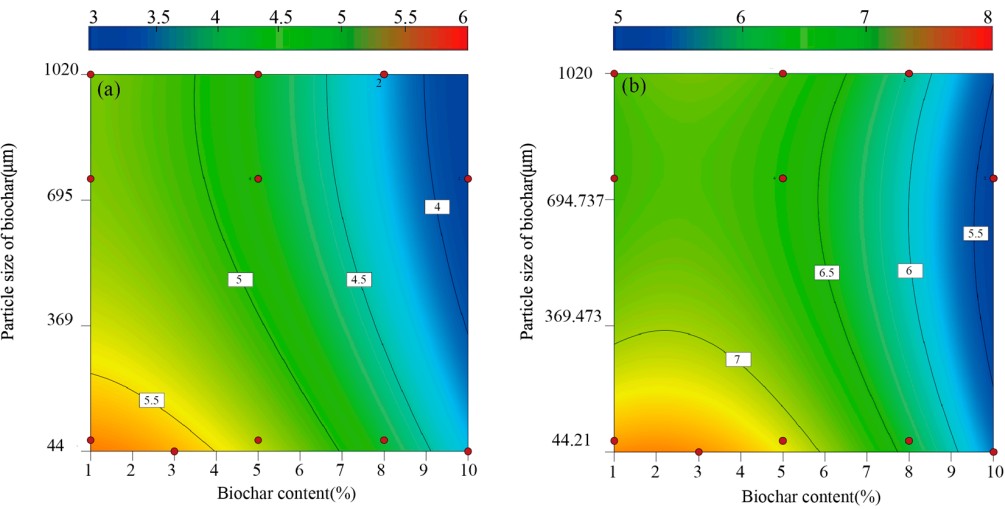

**Figure 5.** RSM analysis of flexural strength: (**a**) 2D plots for 7-day flexural strength; (**b**) 2D plots for 28-day flexural strength (Note: The red dot represents the actual flexural strength value).

### 3.3. Statistical Interpretation of the Test Results

The ANOVA is a powerful tool for quantifying the impact of various factors on the regression models of each response. Typically, the significance of the established model in relation to the response is assessed using an F-value, with the *p*-value as the parameter for hypothesis evaluation. A model is considered insignificant when $p > 0.05$, while $p \leq 0.05$ signifies significance, with a smaller *p*-value indicating higher significance. Tables 6 and 7 show F-values of 23.88, 18.52, 29.72, and 17.61 for the regression models of cement mortar, all with *p*-values below 0.0001. These results confirm the effectiveness of the regression equations in accurately portraying the relationship between cement mortar strength and its influencing factors. Furthermore, it is worth noting that the lack of fit in each model exceeds 0.05, indicating insignificance. The lack of fit in each model is not significant, which means that the models effectively capture the relationship between the experimental factors and the response variable. This also underscores the excellent quality of fit in the response models. Detailed information on the normality of residuals and other relevant data for this analysis can be found in Appendix A.

**Table 6.** The ANOVA results of compressive strength parameters.

| Response | Source | Sum of Squares | df | Mean Square | F-Value | *p*-Value | Remarks |
|---|---|---|---|---|---|---|---|
| $Y_1$ | Model | 17.19 | 5 | 3.44 | 23.88 | <0.0001 | significant |
| | A | 14.09 | 1 | 14.09 | 97.87 | <0.0001 | |
| | B | 0.6006 | 1 | 0.6006 | 4.17 | 0.0683 | |
| | AB | 0.1212 | 1 | 0.1212 | 0.8421 | 0.3804 | |
| | $A^2$ | 1.46 | 1 | 1.46 | 10.17 | 0.0097 | |
| | $B^2$ | 0.4718 | 1 | 0.4718 | 3.28 | 0.1003 | |
| | Residual | 1.44 | 10 | 0.1439 | | | |
| | Lack of Fit | 1.01 | 5 | 0.2015 | 2.33 | 0.1868 | not significant |
| | Pure Error | 0.4317 | 5 | 0.0863 | | | |
| | Cor Total | 18.63 | 15 | | | | |
| $Y_2$ | Model | 28.43 | 5 | 5.69 | 18.52 | <0.0001 | significant |
| | A | 26.15 | 1 | 26.15 | 85.15 | <0.0001 | |
| | B | 1.92 | 1 | 1.92 | 6.26 | 0.0313 | |
| | AB | 0.0583 | 1 | 0.0583 | 0.1899 | 0.6722 | |
| | $A^2$ | 0.3617 | 1 | 0.3617 | 1.18 | 0.3032 | |
| | $B^2$ | 0.0192 | 1 | 0.0192 | 0.0624 | 0.8078 | |
| | Residual | 3.07 | 10 | 0.3070 | | | |
| | Lack of Fit | 2.53 | 5 | 0.5066 | 4.72 | 0.0570 | not significant |
| | Pure Error | 0.5372 | 5 | 0.1074 | | | |
| | Cor Total | 31.50 | 15 | | | | |

**Table 7.** The ANOVA results of the flexural strength parameters.

| Response | Source | Sum of Squares | df | Mean Square | F-Value | *p*-Value | Remarks |
|---|---|---|---|---|---|---|---|
| $Y_3$ | Model | 4.51 | 5 | 0.9013 | 29.72 | <0.0001 | significant |
| | A | 4.27 | 1 | 4.27 | 140.69 | <0.0001 | |
| | B | 0.1493 | 1 | 0.1493 | 4.92 | 0.0508 | |
| | AB | 0.0003 | 1 | 0.0003 | 0.0092 | 0.9255 | |
| | $A^2$ | 0.0276 | 1 | 0.0276 | 0.9089 | 0.3629 | |
| | $B^2$ | 0.0110 | 1 | 0.0110 | 0.3624 | 0.5606 | |
| | Residual | 0.3033 | 10 | 0.0303 | | | |
| | Lack of Fit | 0.2677 | 5 | 0.0535 | 7.54 | 0.0824 | not significant |
| | Pure Error | 0.0355 | 5 | 0.0071 | | | |
| | Cor Total | 4.81 | 15 | | | | |
| $Y_4$ | Model | 5.67 | 5 | 1.13 | 17.61 | 0.0001 | significant |
| | A | 4.49 | 1 | 4.49 | 69.80 | <0.0001 | |
| | B | 0.3065 | 1 | 0.3065 | 4.76 | 0.0541 | |
| | AB | 0.0146 | 1 | 0.0146 | 0.2271 | 0.6439 | |
| | $A^2$ | 0.8133 | 1 | 0.8133 | 12.63 | 0.0052 | |
| | $B^2$ | 0.0449 | 1 | 0.0449 | 0.6980 | 0.4230 | |
| | Residual | 0.6437 | 10 | 0.0644 | | | |
| | Lack of Fit | 0.6433 | 5 | 0.1287 | 1429.47 | 0.6510 | not significant |
| | Pure Error | 0.0005 | 5 | 0.0001 | | | |
| | Cor Total | 6.31 | 15 | | | | |

Furthermore, the coefficient of determination ($R^2$) evaluates the degree of correspondence between experimental results and the fitted model. Values closer to 1 indicate greater reliability for the fitted model. When considering both the adjusted coefficient of determination (Adj.$R^2$) and the predictive coefficient of determination (Pre.$R^2$), a difference of less

than 0.2 suggests the model's strong explanatory power. Additionally, the Adeq Precision gauges the model's ability to withstand disturbances. A higher ratio signifies improved predictive accuracy and greater model reliability. Typically, a value exceeding 4 confirms the model's usefulness.

As shown in Table 8, the $R^2$ values for both compressive and flexural strength models are greater than 0.90. These high $R^2$ values emphasize the robust correspondence between experimental and predictive outcomes. Notably, the difference between the predictive and adjusted $R^2$ values is within 0.2, indicating a strong alignment between the two.

**Table 8.** Response models for compressive and flexural strength parameters of BC-cement mortar.

| Response Parameters | Response Surface Models | Source | $R^2$ | Adj $R^2$ | Pre.$R^2$ | Adeq Precision |
|---|---|---|---|---|---|---|
| Compressive strength | $\text{Sqrt}(Y_1) = 27 - 1.13 \times -0.2826 \times B$ $-0.1468 \times AB - 0.7178 \times A^2 + 0.5194 \times B^2$ | Quadratic | 0.9227 | 0.8841 | 0.8120 | 14.3638 |
| | $\text{Sqrt}(Y_2) = 36.70 - 1.78 \times A - 0.5057 \times B$ $+0.1017 \times AB - 0.33568 \times A^2 - 0.1046 \times B^2$ | Quadratic | 0.9025 | 0.8538 | 0.6962 | 13.3998 |
| Flexural strength | $\text{Sqrt}(Y_3) = 1.58 - 0.1547 \times A - 0.0566 \times B - 0.0090$ $\times AB - 0.0638 \times A^2 + 0.0301 \times B^2$ | Quadratic | 0.9650 | 0.9475 | 0.8604 | 22.0164 |
| | $\text{Sqrt}(Y_4) = 2.57 - 0.1489 \times A - 0.0388 \times B - 0.0228 \times$ $AB - 1.071 \times A^2 +$ $0.0549 \times B^2$ | Quadratic | 0.9512 | 0.9268 | 0.8217 | 19.4376 |

The ultimate model equations for the mortar response, encompassing all model terms, are detailed in Table 8. Based on the analysis provided above, it is evident that these regression models exhibit high fitting accuracy and commendable reliability. Consequently, employing the variance analysis model equations facilitates the estimation of compressive strength and flexural strength in mortar.

*3.4. Optimization and Model Validation*

Using *Design Expert 12.0*, we systematically optimized the mechanical properties of BC cement mortar. The main goal was to precisely determine the optimal BC content and particle size that would yield the highest compressive and flexural strengths within the cement mortar. To accomplish this, we employed a mathematical algorithm based on gradient descent to search for the optimal solution, accounting for various factors' upper and lower limits and practical feasibility. The results indicated that the ideal BC content was determined to be 2.69%, while the optimal particle size was found to be 51.08 μm. Under this combination of factors, both the predicted compressive strength and the predicted tensile strength reach their maximum. Subsequently, we proceeded to prepare a cement mortar specimen in accordance with this optimized scheme and subsequently conducted experimental verification, and the results are presented in Table 9.

**Table 9.** Predicted vs. actual mechanical strength of the optimized mixture.

| Optimum Mix (wt.%) | | Compressive Strength (MPa) | | | | | | Flexural Strength (MPa) | | | | | |
|---|---|---|---|---|---|---|---|---|---|---|---|---|---|
| | | 7 d | | | 28 d | | | 7 d | | | 28 d | | |
| A/% | B/μm | Pred. | Meas. | PE | Pred. | Meas. | PE | Pred. | Meas. | PE | Pred. | Meas. | PE |
| 2.69 | 51.08 | 29.31 | 30.12 | 0.96 | 38.96 | 37.55 | 0.93 | 5.53 | 6.07 | 0.91 | 7.66 | 7.21 | 0.94 |

The data reveal slight discrepancies between the model-predicted compressive strength and the corresponding values measured at 7 days and 28 days, with marginal deviations of 2.7% and 7%, respectively. Similarly, the differences between the projected flexural

strength and the measured values for the 7-day and 28-day periods are merely 9% and 6%, respectively. Notably, all deviations remain well below the 10% threshold. This robustly confirms the logical and dependable nature of the response surface analysis-derived model, emphasizing its significant utility in optimizing the composition of BC-cement mortar with valuable practical implications.

### 3.5. Microstructure Analysis

Figure 6a displays the SEM image of pure cement mortar after 28 days of curing. The image depicts the presence of crystalline calcium hydroxide (CH), ettringite crystals (AFt), and calcium-silicate-hydrate (C-S-H) compounds within the pure cement specimen [7]. These substances combine to form a continuous network structure, which acts as the primary framework for the system. Additionally, it is found that the presence of numerous micropores amidst the reaction product results in a relatively loose structure [16]. Furthermore, cracks are evident in the interfacial transition zone (ITZ), limiting their effectiveness in establishing connections [34].

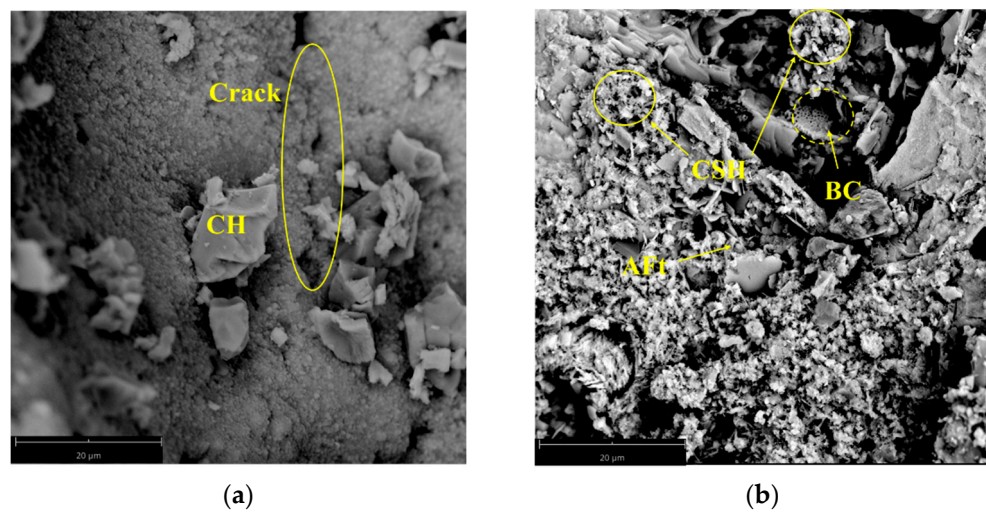

**Figure 6.** SEM images of cement mortar. (**a**) Control group; (**b**) optimal ratio group.

Notably, the inclusion of 2.69% BC in the cement mortar composition significantly enhances the formation of CH, C-S-H, and AFt during the hydration process (Figure 6b), in contrast to the pure cement system. This phenomenon clearly indicates that BC influences the cement hydration reaction. Additionally, it is also observed that there is a discernible presence of BC particles on the surface of the cement mortar. This morphological change underscores the role of finer BC particles as micro-fillers, effectively reducing voids and cracks within the cement matrix and at the interfacial transition zone [25,35]. Consequently, this contributes to the creation of a more compact colloidal matrix. Figure 6b reveals that the pores of the BC particles contain hydration products such as C-S-H. This can be attributed, in part, to the diverse functional groups present in BC particles. This includes carboxyl, hydroxyl, and phenolic groups, which can form stable calcium complexes when interacting with calcium ions in the cement [36]. These complexes serve as additional sources of calcium for the formation of C-S-H gel. Moreover, BC's porous structure supports the growth of C-S-H gel, fostering a more uniform and denser lamellar or fibrous structure. Ultimately, this enhancement contributes to the increased strength of the cement mortar.

## 4. Discussion

The present study aimed to enhance the mechanical properties of cement mortar by incorporating pulverized BC and optimizing its particle size and content using the RSM. Our experimental results highlight that 1.0–5.0% BC is beneficial to improving the mechanical properties of mortars. This improvement is primarily attributed to the role

of BC particles as an internal curing agent and filling effect [37]. Leveraging response surface methodology, we identified the optimal combination of BC particle size and content (51.08 μm and 2.69%), respectively. Under these conditions, the compressive strength and flexural strength of cement mortar reach their optimal values. These findings are consistent with the studies by Tan et al. [38] and Qin et al. [39], who reported that incorporating BC at levels 1–3% can enhance the mechanical properties of cement-based materials.

The analysis of the experimental data reveals a clear correlation between the BC particle size and the mechanical properties of the cement mortar. Specifically, it was observed that finer BC particles led to a noticeable increase in the compressive strength of the mortar when compared to coarser particles. This result can be attributed to the higher surface area provided by finer particles, which facilitates improved interfacial bonding with the cement matrix. This observation concurs broadly with the findings reported by Gupta et al. [40]. Additionally, the finer particles likely contributed to reduced void spaces within the mortar, leading to enhanced density and strength. The study by Ling et al. [25] also proved that the BC particle size of about 40–70 μm has the most significant effect on the mechanical strength of cement-based materials. However, Choi et al. [41] showed that finer BC particles will increase the viscosity of the mortar, resulting in a decrease in fluidity. At the same time, finer BC particles are also easy to float or sink in the cement pastes, resulting in an increase in segregation. This is also proved in the study of Ref [40], that is, 1–5% BC content can reduce the fluidity of cement mortar from 136 mm to 118 mm, respectively. Therefore, when optimizing the mechanical properties of the cement mortar using BC, it becomes necessary to consider both the particle size and the content to achieve a balance between strength enhancement and maintaining the practical fluidity of the mortar.

The findings of this study have significant implications for the practical implementation of BC-enhanced mortar in construction. The findings highlight the potential for enhanced mechanical properties under the interaction of BC content and particle size, providing a route to the development of more sustainable and resilient building materials. Careful consideration of both particle size and content is essential to harnessing the full benefits of BC without compromising fluidity or mechanical performance. To maximize the effectiveness of BC incorporation, researchers should carefully select the appropriate particle size and content based on the specific project requirements and desired outcomes. Furthermore, these results can guide the formulation of guidelines for incorporating BC into mortar mixes, ensuring consistent and reliable results across different applications.

The experiments were carried out in controlled laboratory settings. However, our results are limited by the lack of analysis on the effects of environmental factors and construction scenarios. Other parameters may also affect the results, such as curing conditions and BC source. These variables need more investigation. Research should investigate the durability and sustainability of cement mortar with BC. Additionally, exploring how BC interacts with other sustainable additives, such as recycled aggregates or cementitious materials, can improve construction materials. This avenue of research could lead to innovative solutions for more environmentally friendly and durable building materials.

## 5. Conclusions

This study investigates the impact of substituting 0–10% of cement weight with various particle size levels of BC on cement mortar mechanical properties. Through the utilization of RSM, predictive models for 7-day and 28-day flexural and compressive strengths were derived and compared alongside microstructural analyses involving SEM. The main conclusions are as follows:

(1) The regression models for 7-day and 28-day compressive strength exhibited $R^2$ values of 0.974 and 0.956, respectively. Similarly, the flexural strength regression models for the corresponding time periods demonstrated $R^2$ values of 0.97 and 0.94, respectively. These substantial $R^2$ values, accompanied by the remarkable significance probability levels ($p < 0.0001$), validate the accuracy and effectiveness of the established response surface model. Notably, the response surface analysis underscores that particle size

and content exert a more pronounced influence on the flexural strength of cement mortar compared to compressive strength;

(2) The RSM optimization findings reveal optimal performance with 2.69% BC replacement at a particle size of 51.08 μm, and the absolute relative error between the strength prediction value and the test value is less than 9%, which indicates that the model can provide a reference for the properties optimization of BC mortar;

(3) The SEM experimentation findings provided additional validation of the BC's filling effect and nucleation. Specifically, the BC particles effectively filled the pores within the cementitious materials, with a particular emphasis on the interface transition zone gaps. Furthermore, these BC particles were comprehensively encapsulated by the hydration products, including C-S-H, CH, and AFt, thus facilitating nucleation and promoting the hydration degree of the cement.

**Author Contributions:** Investigation, K.T.; Data curation, Y.C.; Writing—review & editing, Z.Z.; Funding acquisition, J.W. All authors have read and agreed to the published version of the manuscript.

**Funding:** This study was supported by the Young Scholar project of South China University of Technology (000002112124).

**Institutional Review Board Statement:** Not applicable.

**Informed Consent Statement:** Not applicable.

**Data Availability Statement:** Some or all data, models, or code that support the findings of this study are available from the corresponding author upon reasonable request.

**Conflicts of Interest:** The authors declare no conflict of interest.

## Appendix A

Residual analysis plays a crucial role in evaluating the model's adequacy, given its capability to identify systematic biases or deviations from the assumptions inherent in the response model [42]. Figures A1 and A2a show that the residuals follow a mostly linear pattern on the standard probability plot. This observation implies the continued validity of the response model within the defined design space. Moreover, Figures A1 and A2b further illustrate the connection between the residuals and the sequence of runs, clearly demonstrating that they consistently remain within the confines of the designated red control limits. This, in turn, confirms the absence of substantial deviations within the model's processes.

Regression analysis, on the other hand, serves as a method to validate the accuracy of the regression model by comparing the consistency between the model predictions and the actual values. As depicted in Figures A1 and A2c, the comparison between predicted and actual values of compressive strength and flexural strength at different ages shows a distribution close to the y = x line. This implies that the model predictions closely match the actual values, indicating minimal experimental error and a high level of model accuracy.

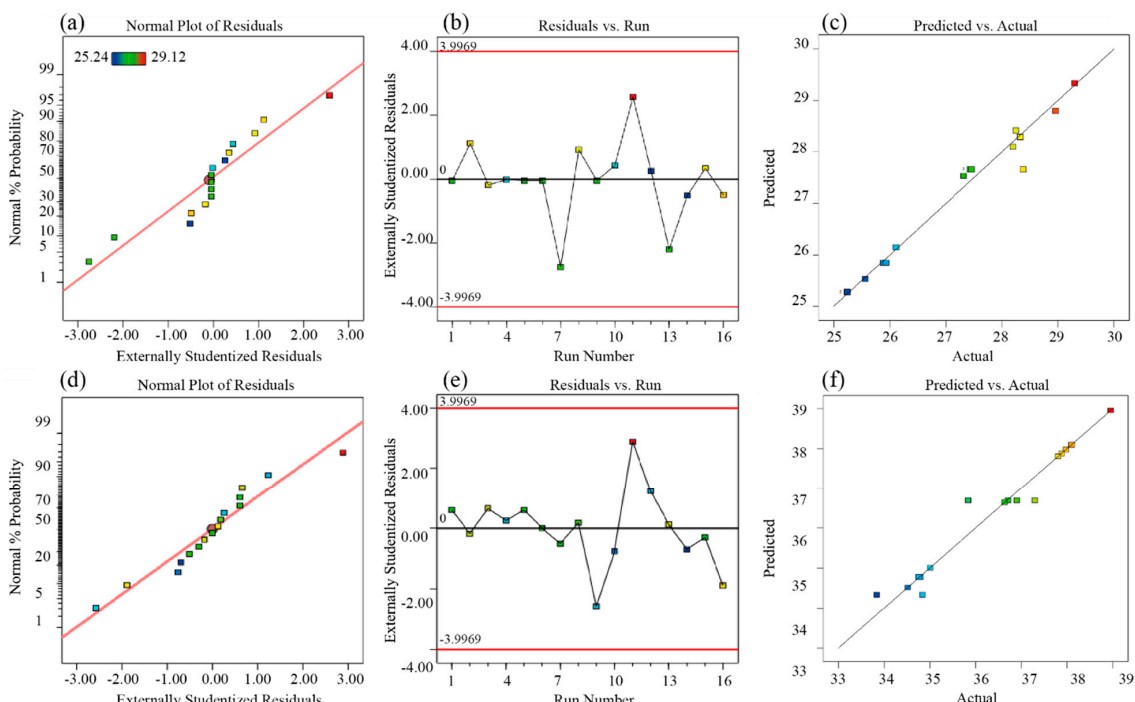

**Figure A1.** Diagnostic procedures for the normal plot of residuals, residual vs. run, and predicted vs. actual compressive strength models. (**a**–**c**) Seven d compressive strength; (**d**–**f**) 28 d compressive strength.

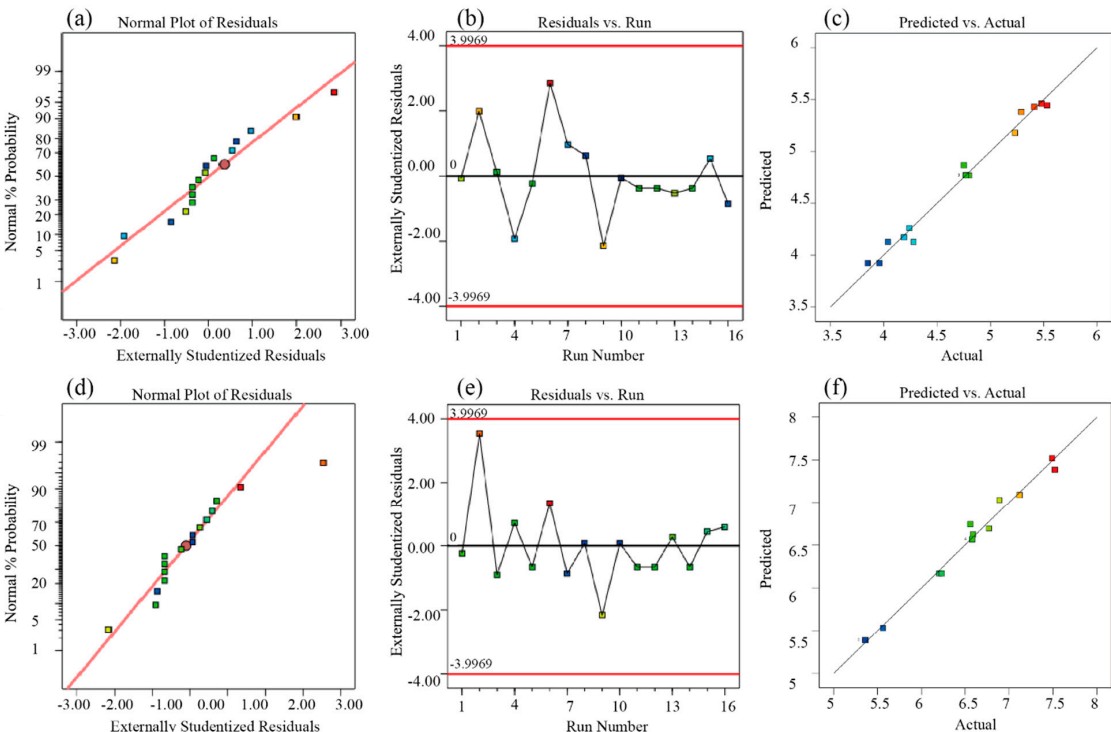

**Figure A2.** Diagnostic procedures for the normal plot of residuals, residual vs. run, and predicted vs. actual flexural strength models. (**a**–**c**) Seven d flexural strength; (**d**–**f**) 28 d flexural strength.

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
