# Peer review of "Enhancing Biochar Impact on the Mechanical Properties of Cement-Based Mortar: An Optimization Study Using Response Surface Methodology for Particle Size and Content"

_sustainability, doi:10.3390/su152014787_

Round 1

Reviewer 1 Report

Table 3 shows particle size variation as 44, 73, 750 and 1020

What was the basis of this variation? why so much gap between 73 and 750?

Figure 5 shows particle size as 44, 369, 694, 1020 which could be a better selection of size parameter, right?

Ultimately, it is shown that 2.6% and 44 size looks good, then this again raises questions on selection of parameters.

BC content should have been chosen like 0.5, 1, 1.5, 2, 2.5%

and particle sixe should have been chosen 10, 25, 40 and 60 micron

How do you explain the wrong selection of parameters?

what is the novelty of this work? Because there are many published articles in this area.

 A lot of work is already done on this topic, there is nothing new

Explain how your work is unique?

Also, standard deviation or scatter bar is not shown in the test data, how many specimens where tested for each case? 

Reviewer 2 Report

The paper presents a topic of interest to the scientific community. The results are presented in a clear way. I only have a few brief comments:

-The novelty of the work should be emphasized.

-There are phrases throughout the manuscript that should be referenced. Particularly in the results section.

-What is the perspective of this investigation?

-Check the template format used. It seems that it is not updated.

None

Reviewer 3 Report

I would like to express my gratitude for the opportunity to review your manuscript “Enhancing Biochar Impact on the Mechanical Properties of Cement-Based Mortar: An Optimization Study Using Response Surface Methodology for Particle Size and Content.” I want to congratulate you on your work; overall, it is very interesting, and the writing is clear and easy to follow. Your study investigates the impact of substituting a certain percentage of cement with biochar (BC) while considering particle size variations on the mechanical properties of cement mortar. The use of Response Surface Methodology (RSM) to analyze the data is commendable.

However, there are several points that need improvement before the manuscript can be considered for publication. I would like you to consider the following observations:

I) Please review the figure captions, as most of them are unclear and do not provide sufficient information to understand the figures. Ensure that figures mentioned in the text are appropriately discussed.

II) Some discussions in the manuscript appear contradictory to the presented results. Please review these discussions to ensure consistency.

III) Some of the conclusions drawn in the manuscript appear speculative, especially regarding the presence of pores in the samples. Please provide more concrete evidence to support these conclusions.

IV) Discuss figures referenced in the text but not included in the manuscript to enable a comprehensive evaluation.

V) Provide more detailed descriptions of the equipment used for testing. Include information about the equipment models and testing conditions.

In terms of the manuscript's structure:

The abstract is concise and clear.

The introduction provides a good context for the study.

The materials and methods section is generally well-described in terms of sample preparation. However, the descriptions of equipment and testing conditions are lacking.

In Figure 2. The legend does not describe the differences between curves a and b. Furthermore, the nomenclature used for the samples as a function of time is not the same as that which appears in the text nor the same as that which is then maintained for the rest of the work. If Fig 2 b is not going to be discussed, it makes no sense for it to be kept in the manuscript. Other points that are not clear in this figure and are not discussed, Why is there an inversion in the pattern of the figure of 20s and 30s, that is, why is the sample with 3s larger than the one with 20s? and then for the 1m and 2m times the curves are practically identical, nothing about this is discussed.

In Figure 3. It is impossible to analyze the micrographs without knowing the size bar. Furthermore, the result seems to be the opposite of what is stated by the authors (considering that they all have the same magnification) the legend of figure 3 is as follows: Figure 3. SEM images of biochar samples with different particle sizes. (a) D50=44.70 μm; (b) D50=73.28 μm; (c) D50=750 125 μm; (d) D50=1020 μm. However when looking at what appears to occur is an increase in small materials from a to d. Which corroborates what is discussed in part of the text. However, without the bars it is impossible to be sure.

Still in these microscopies, the authors state “The surface of BC particles exhibits numerous microscopic pores, which are related to the pore characteristics of the raw wood and the release of volatile gases during pyrolysis. These pores enhance the specific surface area and adsorption capacity of the BC particles. As the mechanical grinding time extends, a substantial quantity of irregular aggregates emerges on the BC particle surface, as shown in Figure 3. These particles exhibit particle dimensions spanning from a few micrometers to tens of micrometers. Consequently, the density of the surface's micropores and specific surface area amplify, thereby furnishing increased adsorption volume. Importantly, this mechanical grinding procedure predominantly affects the BC's macroscopic pore arrangement, rendering its particle morphology erratic with sharp edges.” However, in the images presented, it is impossible to visualize any pores in the samples. In this excerpt it is clear that the authors attribute the appearance of smaller materials to grinding as time increases, which corroborates the surface area values presented in table 2. Now what is contradictory to the nomenclature presented here is that then follows all the work in which the BC has the smallest (D50) specific area while the largest BC (D50) has the largest specific area. I suggest that it be reviewed because something doesn't make sense either here or in the rest of the work.

The authors mention pores a lot and talk about adsorption without showing the isotherms, they only present the area values in Table 2, but they don't even discuss anything or even mention these results in the text.

Throughout the text the authors mention the software used “Design Expert 9.0” and never once does it appear in italics.

In the experimental design, please justify the use of a custom design method rather than a full factorial design, considering the number of factors and levels. Clarify how you handled alias structures in the confusion matrix.

Furthermore, it is not clear in the text or in table 5 why test 10 was carried out with a concentration of 8%? Another questionable point about this table is that in the methodology the text mentions that these tests were carried out in triplicates, but the table does not mention whether the results discussed are average and, if so, the standard deviation of the results is not presented. This is very serious because it becomes impossible to know whether the variations observed are really significant, as discussed by the authors (it should be noted that the equipment on which the tests were carried out was not mentioned, nor was its accuracy).

Next, Figure 4 is presented, again the figure caption is not clear about what figures (a) and (b) are. But even more serious is the way in which they are presented simply as a plot of the data presented in table 5, that is, the results are presented as in the experiments in a random way, which makes it practically impossible to interpret them in relation to the factors evaluated. Furthermore, the lack of a standard deviation bar and also the data from the control sample make it impossible to see how significant the data presented is.

The discussion of Experiment 12 should consider the possibility of it being an outlier due to the unique 3% concentration.

In Figure 5, clarify what each part (a-d) represents. It appears that only Figure 5(a) is discussed, rendering the others seemingly irrelevant.

In the discussion “Maintaining a constant particle size for biochar while varying its content within the range of 1% to 5%, the compressive strength of the cement mortar specimens exhibits a minor yet discernible enhancement.” This discussion is clearly mistaken. When observing the curves generated, it is clear that with the increase in concentration for all samples there is a decrease in the “compressive strength”.

Next, Figure 6 is presented, which presents the same problem in the caption as the previous ones. But your discussion is perhaps the best of the work. In this case, only figures (a) and (b) are discussed in the text.

Points that need to be reviewed:

i) The models that gave rise to the curves are not presented, nor is anything discussed about them.

ii) Clearly the RSM indicate a tendency towards higher values for the region with smaller fiber size and concentration, could this be a bias in the model? The authors at no point justify this, an interpretation of the curves alone without considering the data in the table that indicate that the control has a lower value would lead to the assumption that the addition of any quantity and size of BC reduces the “compressive strength”.

iii) Another point that is not discussed is what the red and bank dots are on the contour lines. At no point do the authors discuss what these points are.

The models presented in the ANOVA tables are not discussed in the manuscript. Please provide explanations for these models, and consider presenting the table earlier in the text.

When discussing the optimization of mechanical properties, provide more details on how this was achieved, the parameters considered, and the theory behind the software's conclusions.

Figure 9 is presented, and when reading the text, a discussion is made about figures 9 (c - d), which are not present in the text.

The authors then say “. Furthermore, the analysis of the experimental data reveals a clear correlation between BC particle size and mechanical properties. Finer BC particles demonstrated a notable increase in the compressive strength of the mortar compared to coarser particles. This result can be attributed to the higher surface area provided by finer particles, which facilitates improved interfacial bonding with the cement matrix.”

However, if we remember the data presented in figure 3, this is contradictory to the notation used since the sample that the authors call D50=1050 has the largest fiber, but in microscopy it is clearly the one with the smallest particles and the largest specific area.

I appreciate the effort you've put into this research, and I believe that addressing these points will significantly improve the quality and clarity of your manuscript.

Best regards.

Round 2

Reviewer 1 Report

Looks ok but not completely satisfied by answers

Author Response

We have revisited our manuscript in light of the comments made by the reviewers and revised it accordingly. Please see the attachment

Reviewer 3 Report

I have had the opportunity to review the manuscript titled "Enhancing Biochar Impact on the Mechanical Properties of Cement-Based Mortar: An Optimization Study Using Response Surface Methodology for Particle Size and Content" submitted for publication in Sustainability. After authors revision feedback, I careful consideration, I am pleased to recommend the publication of this article with only minimal revisions. The manuscript presents a well-structured and comprehensive exploration of the subject matter. The research conducted is rigorous, and the methodology employed is sound. The findings are of significant importance to the field and contribute valuable insights that advance our understanding of cement additive with biochar. In terms of revisions, there are a few minor points that should be addressed before publication. These include:

Some figures continue to appear in the manuscript but are not referenced in the text. I strongly suggest that the authors choose to reference the figures in the text and associate each of them with the corresponding discussion. For example, in line 128, "The SEM images of BC particles at various magnifications are shown in Figure 2." The authors initiate the discussion of Figure 2; however, Figure 2 is divided into 2(a) and 2(b). It is crucial that this distinction is made in the text, and observations about each figure are discussed clearly. The same applies to Figures 3(a) and 3(b).

Regarding Figures 4(b), 4(c), 4(d), and 5(c), 5(d), they are neither mentioned nor discussed in the text. If the authors choose not to reference or discuss them, it would be advisable to remove them from the manuscript.

Furthermore, the text mentions Figures 6, 7, and 8, but their captions are incorrect (they are labeled as Figures 4, 5, and 6).

Regarding the response to comment 15, "As for the ANOVA tables, do you mean we need to put them prior to which part of the paper?" I suggest that they be placed at the beginning of the statistical discussion. In particular, I would include the analysis of assumptions (normality of residuals and other relevant data from this section) in a supplementary material, as these are merely conditions that validate the models to be presented. The discussion accompanying the ANOVA, on the other hand, is crucial as it demonstrates the contributions of the models.

Furthermore, I would like to extend my congratulations for the excellent work and the efforts made to enhance its quality.

I believe that addressing these minor points will enhance the overall quality of the article and ensure that it meets the high standards of Sustainability.
